# Reducing Time and Computing Costs in EC-Earth:
# An Automatic Load-Balancing Approach for Coupled ESMs

Sergi Palomas, Mario C. Acosta, Gladys Utrera, and Etienne Tourigny

Barcelona Supercomputing Center, Plaça d'Eusebi Güell, 1-3, 08034 Barcelona, Spain

**Correspondence:** Sergi Palomas (sergi.palomas@bsc.es)

**Abstract.** Earth System Models (ESMs) are intricate models employed for simulating the Earth's climate, typically constructed from distinct independent components dedicated to simulate specific natural phenomena (such as atmosphere and ocean dynamics, atmospheric chemistry, land and ocean biosphere, etc.). In order to capture the interactions between these processes, ESMs utilize coupling libraries, which oversee the synchronization and field exchanges among independent developed codes

typically operating in parallel as a Multi Program, Multiple Data (MPMD) application.

The performance achieved depends on the coupling approach, as well as on the number of parallel resources and scalability properties of each component. Determining the appropriate number of resources to use for each component in coupled ESMs is crucial for efficient utilization of the High Performance Computing (HPC) infrastructures used in climate modelling. However, this task traditionally involves manual testing of multiple process allocations by trial and error, requiring significant time in-

vestment from researchers. Thus, making the process more error-prone, and often resulting in a loss in application performance due to the complexity of the task. This paper introduces the automatic load-balance tool (auto-lb), a methodology and tool for determining the resource allocation to each component within coupled ESMs, aimed at improving the application's performance. Notably, this methodology is automatic and does not require expertise in HPC to improve the performance achieved by coupled ESMs. This is accomplished by minimizing the load-imbalance: reducing each constituent's execution cost (core-

15   hours), as well as minimizing the core-hours wasted resulting from the synchronizations between them, without penalizing the execution speed of the entire model. This optimization is achieved regardless of the scalability properties of each constituent and the complexity of their dependencies during the coupling.

To achieve this, we designed a new performance metric called "Fittingness" to assess the performance of coupled execution evaluating the trade-off between the parallel efficiency and application throughput. This metric is intended for scenarios where

optimality can depend on various criteria and constraints. Aiming for maximum speed might not be desirable if it leads to a decrease in parallel efficiency and, therefore, increasing the computational costs of simulation.

The methodology was tested across multiple experiments using the widely recognized European ESM, EC-Earth3. The results were compared with real operational configurations, such as those used for the Coupled Model Intercomparison Project Phase 6 (CMIP6) and European Climate Prediction project (EUCP), and validated on different HPC platforms. All of them

suggest that the current approaches lead to performance loss, and that auto-lb can achieve better results in both, execution speed and reduction of the core-hours needed. When comparing to the EC-Earth standard-resolution CPMIP6 runs, we achieved a configuration 4.7% faster while also reducing the core-hours required by 1.3%. Likewise, when compared to the EC-Earth

high-resolution EUCP runs, the method presented showed an improvement of 34% in the speed, with a 6.7% reduction in the core-hours consumed.

*Copyright statement.* TEXT

## 1 Introduction

In the field of climate science, the adoption of advanced modelling techniques has become imperative for understanding and predicting the complex dynamics of our planet's climate system. The recognition of the complex interconnectedness among various natural phenomena, crucial for describing the climate, led to the development of Coupled General Circulation Models

(CGCMs) more than 40 years ago, as illustrated by Manabe et al. (1975). These models captured the physical processes occurring in both the atmosphere and ocean. To further represent the natural feedback loops and avoid using predefined data on the given region by boundary conditions led to the creation of Earth System Models (ESMs), which seek to simulate all relevant aspects of the Earth system, expanding the limits of CGCMs by simulating carbon cycle, aerosols, and other chemical and biological processes (Valcke et al., 2012; Lieber and Wolke, 2008). Consequently, coupling multiple codes that simulate

different natural phenomena has become an common practice to better represent the climate.

Various strategies exist for designing the coupling approach in ESMs. Frequently, multiple independently developed codes run simultaneously and synchronize during the runtime to exchange fields with one another. These applications are commonly referred to as Multi Program, Multiple Data (MPMD), and components running in parallel can employ different parallel paradigms such as Message Passing Interface (MPI) (Tintó Prims et al., 2019) to take advantage of the High Performance

Computing (HPC) machines.

Achieving a "satisfactory" performance on coupled ESMs is challenging, given the inherent complexities of such applications, but also of upmost importance to maximize the number of simulations and the resolution available to the climate research community, while using HPC infrastructures more efficiently. Balaji (2015) showed that current ESMs performance is deteriorated due to the need of coupling. Acosta et al. (2023a) showed in a collection of performance metrics from multiple Coupled

Model Intercomparison Project Phase 6 (CMIP6) experiments that the coupling cost adds, in average, a computational overhead (in core-hours) of 13%. As illustrated in Figure 1, coordination among components is required to exchange the coupling fields, typically utilizing MPI. This often results in faster components waiting for the slower ones, a problem known as the load imbalance. Moreover, extra computation is needed to transform the data between components using different grids, a process known as interpolation. This process, along whit the associated MPI communications, has been studied extensively by Donners

et al. (2012) to evaluate both the efficiency of the interpolation algorithm and the impact of these communications on overall performance. Minimizing the cost associated with the load-imbalance by finding the appropriate resource configuration is a non-trivial process, which includes analysing the speed-up of individual components at various processor counts, study their

interactions during the coupling, and making trade-offs between the computing cost (core-hours) and execution time of the coupled ESM.

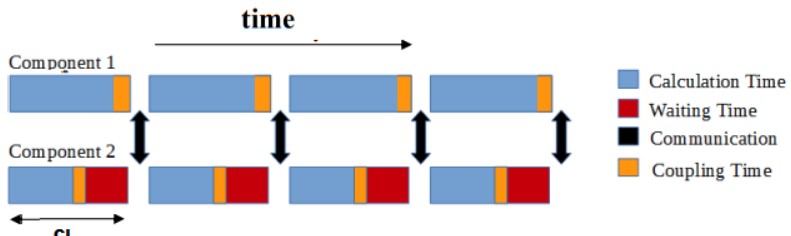

**Figure 1.** Overview of the typical timeline pattern observed between two coupled components during execution. Component 2 exhibits a faster computational time (depicted in blue) than Component 1, leading to Component 2 waiting at the end of each Coupling Interval (CI) (depicted in red). The figure also illustrates the extension of the entire execution due to coupling time (depicted in orange). This typically includes tasks such as regridding and additional calculations necessary before communicating fields across different components

The strategies for load-balancing ESMs can be divided in *dynamic* load balancing, where the load-imbalance is minimised during the runtime, and *static* load-balancing, where the process involves stopping and rerunning the model execution to find resource configurations that minimise the coupling cost. To deal with the load-balance "dynamically" the applications must allow reallocating the processes on which it runs during runtime, a property known as malleability (Feitelson and Rudolph, 1995). Some examples of using checkpoints during the execution have been shown by Vadhiyar and Dongarra (2003); Maghraoui
et al. (2005, 2007). Possibly the most notable contributions to *dynamic* approaches have been done by Kim et al. (2011) to extend the Model Coupling Toolkit (MCT) to Malleable Model Coupling Toolkit (MMCT), enabling malleability and incorporating a load-balance manager module. This module decomposes the time of each component during a Coupled Interval (CI) into *constituent computation* and *constituent coupling*. The load-balance manager reallocates Processing Elements (PEs) from the fastest (donor) to the slowest (recipient) component until solution improvement ceases. This work was further enhanced in
Kim et al. (2012b), where MMCT was extended with a prediction mechanism that maintains a database of PEs-execution times at each iteration, and a manually-generated heuristic optimisation to determine new resource configurations that reduce the coupling step execution time. Kim et al. (2012a) extended this approach to handle applications which have varying workloads during the execution.

However, the manually-generated heuristic used for the prediction -based on static and manual inspection of coupled model
interaction patterns and constituents' computations- becomes impractical for more complex, realistic coupled models. To address this, Kim et al. (2013) proposed an instrumentation-based approach that collects runtime data from the constituents, demonstrating how this information can be used to improve coupling performance and accelerate the load-balancing decision-making process.

While these approaches have demonstrated significant improvements, they are designed for a highly flexible coupling scheme
applicable only to climate models that adopt the MMCT extension of MCT. As a result, they are not suitable for most state-

of-the-art ESMs. Moreover, the method proposed has not been validated with production ESMs used in climate research, but rather with a simplified "toy" model that mimics a simulation of the Community Earth System Model (CESM).

In contrast, our proposed solution is not integrated into any specific coupler, making it readily accessible to most ESMs used by the climate research community that employs an external coupling library to link multiple binaries (MPMD) into a single application.

It is essential to note that all this dynamic load-balancing methods rely on the capacity of adjusting the number of processes a constituent uses during the runtime, a feature which is rarely seen in state-of-the-art ESMs. Additionally, the method testing has been confined to toy models, lacking validation on ESMs widely employed within the scientific community. These limitations underscore the need for further research and adaptation to real-world, complex scientific applications. Furthermore, one can argue that this solutions are not fully "*dynamic*" as suggested, given that the simulation has to be continuously interrupted during the runtime to collect the performance metrics, execute an algorithm to find a better setup, and resuming the simulation. A truly *dynamic* approach should instead have other means to balance the workload to minimise the IDLE time such as tasks, an option explored with the Dynamic Load Balance library developed at the Barcelona Supercomputing Centre (BSC). Garcia et al. (2009); Marta et al. (2012) have explored the possibility to use this tool to reassign computation resources of blocked processes to more loaded ones to speed-up hybrid MPI+OpenMP and MPI+SMPSs applications. Although this is a promising option for the future, the current state of the tool has still room for improvement and thread-level parallelism is not common in the current generation of ESMs.

*Static* load-balancing solutions are well suited for the climate science community due to the difficulties found in effectively applying dynamic approaches. One of the most significant contributions of static load-balancing is the work by Ding et al. (2019, 2014) for CESM, which introduced an auto-tuning component integrated into the CESM framework to optimize process layout and reduce model runtime. It achieves this by employing a depth-first search (DFS) method with a branch-and-bound algorithm to solve a Mixed Integer Nonlinear Programming (MINLP) problem, combined with a performance model of the model components to minimise search overhead. This approach improves upon the earlier method described by Alexeev et al. (2014), which relied on a heuristic branch-and-bound algorithm and a less accurate performance model. Later, Balaprakash et al. (2015) proposed a static, machine-learning-based load-balancing approach to find high-quality parameter configurations for load balancing the ice component (CICE) of CESM. The method involves fitting a surrogate model to a limited set of load-balancing configurations and their corresponding runtimes. This model is then used to efficiently explore the parameter space and identify high-quality configurations. Their approach had to take into account the six key parameters that influence CICE component performance: the maximum number of CICE blocks and the block sizes in the first and second horizontal dimensions (x, y); two categorical parameters that define the decomposition strategy; and one binary parameter that determines whether the code runs with or without a halo. They demonstrated that their approach required 6x fewer evaluations to identify optimal load-balancing configurations compared to traditional expert-driven methods for exploring feasible parameter configurations.

Importantly, coupling in CESM follows an *integrated coupling framework* strategy (Mechoso et al., 2021), where the climate system is divided into component models that function as subroutines within a single executable and orchestrated by a coupler

main program (CPL7), which coordinates the interaction and time evolution of the component models. The coupler also allows for flexible execution layouts, enabling components to run sequentially, concurrently, or in a hybrid sequential/concurrent mode. This coupling strategy differs with approaches that use an external coupler (such as OASIS, MCT, or YAC), where existing model codes are minimally modified to interface with the coupling library and executed as separate binaries on different physical cores, either interleaved or concurrently. Furthermore, the performance model used requires generating and analysing execution traces to characterize the computation and communication patterns of key kernels for each coupled component separately. While this can provide more accurate performance representations, it also introduces significant challenges in adapting the approach to new components or other ESMs.

Other *static* solutions, such as those proposed by Will et al. (2017) for the COSMO-CLM regional climate model and Dennis et al. (2012) for CESM, demonstrate that load balancing in widely used ESMs can be approached in a relatively simple manner. These methods aim to identify a resource configuration where all individual components run at roughly the same speed, often constrained by a predefined parallel efficiency threshold. However, as we will show, this approach can easily lead to suboptimal solutions.

In this work, we present a *static* load-balancing method, the automatic load-balance (auto-lb), designed to improve resource allocation in coupled ESMs. Our approach is suited for coupled models that do not support malleability, where each component runs on separate cores as an MPMD application. Unlike other methods, our approach completely eliminates the need to modify any of the component's source codes; instead, it achieves load-balance by adjusting the allocation of PEs assigned to each component. To accomplish this, we have introduce two new performance metrics: firstly, the Partial Coupling Cost to quantify the cost of the coupling per component, and secondly, the Fittingness metric to better address the Energy-To-Solution (ETS, i.e. minimise the energy consumption) and Time-To-Solution (TTS, i.e. minimise the execution time) trade-off prevalent in all non-perfectly scalable applications (Abdulsalam et al., 2015). These advancements set our method apart from existing approaches that either focus exclusively on minimizing execution time (pure TTS) or enforce parallel efficiency thresholds that limit speed in an arbitrary manner.

Moreover, the method includes a prediction phase capable of accurately estimating coupling performance based solely on the scalability properties of the individual model components. Unlike prior work, this eliminates the need for instrumenting the code, using profiling software, or trace generation. The results of the prediction phase allow our approach to significantly reduce the number of real simulations (and thus computational resources) required to determine an improved load-balancing configuration.

Finally, the method is fully integrated in a workflow manager, ensuring that the process of identifying the best resource configuration requires minimal user intervention and aligns with standard practices in climate modelling.

Our research primarily focuses on optimizing real experiment configurations for one of the most prominent European Earth System Model (ESM), EC-Earth3 (Döscher et al., 2022). Notably, EC-Eaerth3 employs the OASIS-MCT coupler (Craig et al., 2017), a widely used coupler also adopted by numerous other ESMs specially in Europe. The new methodology has been used to optimize configurations for different resolutions of EC-Earth3 experiment, including the same experiment configuration used for the CMIP6 exercise (Eyring et al., 2016), and the results of balancing an European Climate Prediction project (EUCP)

high-resolution experiment on the European Centre for Medium-Range Weather Forecasts (ECMWF) CCA machine. This demonstrates that the method can be used across different machines and for different model resolutions, and of its potential applicability to a wide range of ESMs. The method has proven effective, yielding resource configurations that outperform the previous configurations in both execution time and computing cost. As detailed in Section 5, when compared to setup used by the standard-resolution EC-Earth3 CMIP6 runs, we identified a new resource allocation that runs 4.7% faster while reducing the core-hours consumed by 1.3%. Moreover, compared to the performance achieved by the EC-Earth3 high-resolution configuration used in EUCP, we achieved to reduce the execution speed by up to 34%, with a 6.7% reduction in the core-hours needed.

## 2 ESM under study: EC-Earth3

EC-Earth3 is a global coupled climate model developed by a consortium of European research institutions that integrates multiple component models to simulate the Earth system. Its goal is to build a fully coupled atmosphere-ocean-land-biosphere model usable for problems encompassing from seasonal-to-decadal climate prediction to climate change projections and paleoclimate simulations. Fig. 2 shows an overview of the most commonly used EC-Earth3 configuration, EC-Earth3 at standard resolution (EC-Earth3 SR), which couples the ocean (NEMO), the atmosphere (IFS), and the runoff (RNF) components via the OASIS3-MCT coupler. In addition, a parallel IO server (XIOS) is used to better handle the output of the oceanic component. A brief description of the components is listed below:

- The OASIS3-MCT coupler: a coupling library to be linked to the component models and whose main function is to interpolate and exchange the coupling fields between them to form a coupled system.

- The Integrated Forecasting System (IFS) as atmosphere model: an operational global meteorological forecasting model developed and maintained by the European Centre of Medium-Range Weather Forecasts (ECMWF). The dynamical core of IFS is hydrostatic, two-time-level, semi-implicit, semi-Lagrangian and applies spectral transforms between grid-point space and spectral space. In the vertical the model is discretised using a finite-element scheme. A reduced Gaussian grid is used in the horizontal.

- The Nucleus for European Modelling of the Ocean (NEMO) as ocean model: a state-of-the-art modelling framework for oceanographic research, operational oceanography seasonal forecast and climate studies. It discretises the 3D Navier-Stokes equations, being a finite difference, hydrostatic, primitive equation model, with a free sea surface and a non-linear equation of state in the Jackett. The ocean general circulation model (OGCM) is OPA (Océan Parallélisé), a primitive equation model which is numerically solved in a global ocean curvilinear grid known as ORCA. EC-Earth 3.3.2 uses NEMO's version 3.6 with XML Input Output Server (XIOS) version 2.0, an asynchronous input/output server used to minimise previous I/O problems.

- The Louvain-la-Neuve sea-Ice Model 3 (LIM3): a thermodynamic-dynamic sea-ice model directly coupled with OPA.

- The Runoff-mapper (RNF) component: used to distribute the runoff from land to the ocean through rivers. It runs using its own binary and coupled through OASIS3-MCT.

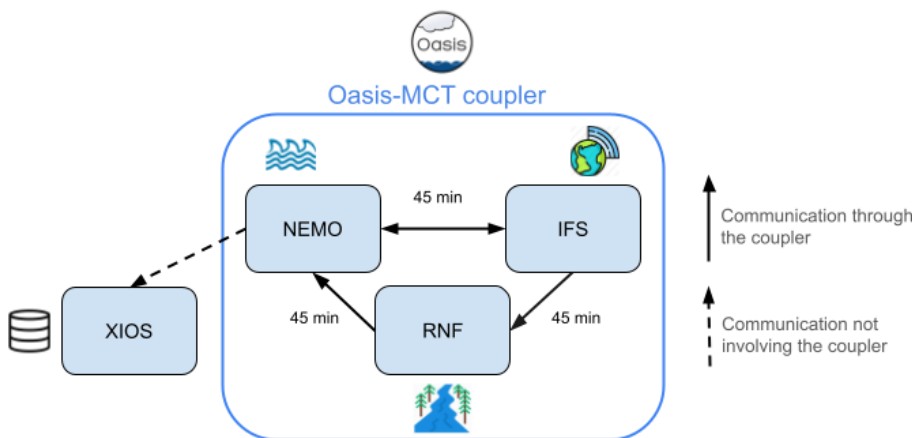

**Figure 2.** Overview of an EC-Earth3 experiment using the Nucleus for European Modelling of the Ocean (NEMO) as the ocean (with sea ice and ocean biogeochemistry), the Integrated Forecasting System (IFS) as the atmosphere, and the Runoff-mapper (RNF) c as the runoff from land to the ocean. Furthermore, we include the XML-IO-Server (XIOS) component, which is used by NEMO to provide asynchronous and parallel IO operations. The arrows show the dependencies between components and the frequency of these interactions in simulated time. Note that XIOS does not communicate through OASIS-MCT. The coupling frequency depicted (45 minutes) corresponds to the Standard Resolution configuration (T255-ORCA1). Higher resolutions use higher coupling frequencies (e.g. 15 minutes for T511-ORCA025)

## 2.1 Experiment configurations

The configurations under study are the Standard Resolution (SR) and High Resolution (HR) simulations (Döscher et al., 2022). They are the most used on EC-Earth3 and, therefore, the ones that consume more computing resources and for which any gain in performance has a greater impact. They both include IFS coupled with NEMO as the main components, parallelized using MPI, and which interchange 23 fields (6 from NEMO to IFS and 17 from IFS to NEMO) through OASIS3-MCT at the beginning of their own timestep. As a consequence, the two components have to be synchronized before starting executing
their own computation.

In SR, IFS uses the T255L91 grid, which corresponds to a resolution of 80 km for the atmosphere, coupled to NEMO using an ORCA1L75 grid, which corresponds to a 1-degree resolution at the equator, or ∼25km (Döscher et al., 2022). In HR configurations, the grids are T511L91 for the atmosphere and ORCA025 for the ocean, which correspond to a resolution of 40 km and $1/4$ of a degree for IFS and NEMO, respectively (Haarsma et al., 2020). They both involve, in addition to NEMO and
IFS, the RNF and XIOS components. For the load-balancing, XIOS and RNF are not taken into account, as XIOS does not

communicate via OASIS but directly with NEMO to handle its IO operations in parallel, and RNF runs in serial and is much faster than the other components.

## 3 Coupled ESMs performance

ESMs are not an exception when it comes to their scaling properties: the parallel efficiency cannot be maintained as we increase the number of PEs used. Thus, boldly selecting the configuration which maximizes the speed leads to a waste of computing resources and is usually avoided. Consequently, efficiency metrics are used to evaluate how the execution cost (i.e. core-hours) increases when adding more resources to a non-perfect scalable model. In other words, how the speed-up of the application responds to the increase of parallel resources for a fixed problem size. Therefore, selecting the appropriate number of PEs to execute the program becomes a trade-off between the speed (Time-To-Solution, TTS) and the core-hours (Energy-To-Solution, ETS) required for the execution, and the proper decision can vary depending on the context: computing resources available, HPC infrastructure policies, scheduling configurations and constraints, and urgency for getting the results.

As seen by Acosta et al. (2023a), another key factor that further deteriorates computational performance in current ESMs is the coupling between their components. This overhead stems from faster components having to wait for slower ones during synchronization, a phenomenon known as load imbalance, as well as the additional computation required to interpolate data between components operating on different grids.

Previous work by Acosta et al. (2023b) has studied this in the context of the EC-Earth3 model, showing that while interpolation process adds to the coupling cost, most of the overhead comes from synchronization delays. Minimizing these costs is crucial to improving the overall performance of the coupled system. However, reducing load imbalance by optimizing resource allocation across components is a complex task. It requires compromising on the parallelisation of individual components to minimise the waiting time during synchronisations. In doing so, we limit the ability to freely choose the best resource configuration for each component, which means some parallel efficiency is lost on the individual components due to not using their best scalability point, but rather the one that bests suits the whole ESM.

This section introduces the performance metrics used during our work to assess the performance of coupled ESMs, as well as presenting both, the problem and adopted solution for the Energy-To-Solution / Time-To-Solution trade-off.

### 3.1 Performance metrics

On the one hand, there are very well-known Speed-up and the Parallel efficiency metrics. Widespread metrics used to assess the performance achieved while dealing with the same amount of work but with different processor counts (scalability with fixed problem size). Given that some of the ESM components cannot run in a single process (serial execution) due to memory and/or computing requirements, the execution in a single node per component ($P_o$) is taken as the baseline instead. Therefore, the Speed-up at $p$ processors is defined as:

$$Speedup_p = \frac{T_{p_o}}{T_p} \qquad (1)$$

Where $T_p$ is the execution time using $p$ processes.

Likewise, the parallel efficiency at $p$ processes is defined as:

$$Efficiency_p = \frac{Speedup_p}{\frac{p}{p_o}} \tag{2}$$

On the other hand, we use a subset of the performance metrics specially designed for the common structure of ESMs and how they are executed in production: the Computational Performance Model Intercomparison Project (CPMIP)(Balaji et al. (2017)). The ones of particular interest for our analysis are listed below:

- **Simulated Years Per Day of execution (SYPD)**: The number of simulated years (SY) by the ESM within a single execution day, defined as 24 hours of computation time on the HPC platform.

- **Core-Hours per Simulated Year (CHSY)**: The core hours per simulated year. Measured as the product of the model runtime for 1 SY (in hours) and the number of cores allocated ($P_M$). Note that the CHSY and SYPD are related by the following formula:

$$CHSY = \frac{24 \cdot P_M}{SYPD} \tag{3}$$

- **Coupling cost (Cpl_cost)**: Measures the overhead caused by the coupling. This can be due to the waiting time caused by

the synchronization between models within the ESM (faster components have to wait for slower ones), the added cost of interpolating the data from the source grid to the target one and the time spent in communications when sending/receiving the data (see Figure 1).

$$Cost = \frac{T_M P_M - \sum_c T_C P_C}{T_M P_M} \tag{4}$$

Where $T_M$ and $P_M$ are the runtime and parallelization for the whole model, and $T_C$ and $P_C$ the same for each component.

For this work, the Eq. 4 has been reformulated to evaluate how much each component adds to the coupling cost, which is essential to know which component should lend PEs, and which one should receive them. It has been called Partial coupling cost:

$$Partial\_cpl\_cost = \frac{T_{Ccpl} P_C}{T_M P_M} \tag{5}$$

Where $T_{Ccpl}$ is the total time spent by a component in coupled events (waiting, interpolating and sending).

All these metrics are collected after the simulation using runtime timing information provided by the load balance tool integrated in OASIS3-MCT (Maisonnave et al., 2020).

## 3.2 Time-to-Solution vs Energy-to-Solution criteria

If we want an application to run faster, we will increase the number of PEs. Assuming that the parallel efficiency decreases (due to non-perfect scalability), the core-hours consumed by the application will rise. Given that the core-hours are directly proportional to the energy cost of execution, they directly influence energy consumption (Balaji et al., 2017). This is known in the literature as the Time-to-Solution (TTS) vs Energy-to-Solution (ETS) trade-off (Freeh et al., 2005).

One of the most commonly used metrics for assessing program performance, which considers both execution time and the parallel efficiency, is the Energy-Delay Product (EDP). In the context of MPI applications, the EDP can be computed as follows (Yepes-Arbós et al., 2016; Abdulsalam et al., 2015):

$$EDP = Speedup \cdot Efficiency \tag{6}$$

In this study, we introduce a novel metric termed "Fittingness metrics" (FN) that enables the parameterisation of the Time-Energy tradeoff at which a program is intended to operate. This metric serves as a valuable tool for assessing and optimizing program performance by considering the balance between execution time and energy consumption. To that end, we define initially two parameters: Time-To-Solution weight ($TTS_w$) and the Energy-To-Solution weight ($ETS_w$). Both parameters are constrained to a range between 0 and 1, and their sum must equal unity:

$$TTS_w + ETS_w = 1 \tag{7}$$

Then, given the scalability curve of one component, with the SYPD (metric of execution time) and CHSY (metric of execution cost) at different core counts, the FN is calculated as follows:

$$FN = TTS_w \cdot SYPD_n + ETS_w \cdot (1 - CHSY_n) \tag{8}$$

Where $SYPD_n$ ($CHSY_n$) is the value of the SYPD (CHSY) after a min-max normalization, which is performed across all tested configurations. Note that we use $1 - CHSY_n$ given that the greater the cost, the less energy efficient the execution will be. In other words, lower costs correspond to improved energy efficiency during execution. Consequently, minimizing the cost not only enhances performance but also reduces core-hours consumption.

Weighting the SYPD (CHSY) with the $TTS_w$ ($ETS_w$) enhances the flexibility in determining the resource configuration that better suit the specific requirements of climate scientists.

Tab. 1 shows how the FN metric compares to the EDP across different $TTS_w$ for the atmospheric component (IFS) in SR. The $SYPD_n$ ($CHSY_n$) column is the value of the SYPD (CHSY) after a min_max normalization. For instance, using 48 PEs for IFS is the slowest configuration ($SYPD_n$=0) but the one that consumes less energy ($1 - CHSY_n$=1). On the other hand, using 1008 PEs is the fastest configuration ($SYPD_n$=1) but the worst in terms of energy ($1 - CHSY_n$=0).

## 3.3 Model performance stability and measurements uncertainty

Evaluating the performance of ESMs inherently involves uncertainty due to the variability of HPC environments. Under an ideal scenario, repeated runs of the same model setup should yield identical runtimes. However, in practice, HPC systems

| | | | | | FN results using different $TTS_w$ | | | | | | |
|---|---|---|---|---|---|---|---|---|---|---|---|
| nproc | $SYPD_n$ | $1-CHSY_n$ | Eff | EDP | ≤0.3 | 0.4 | 0.5 | 0.6 | 0.7 | 0.8 | ≥0.9 |
| 48 | 0.000 | 1.000 | 1.000 | 0.000 | 0.700 | 0.600 | 0.500 | 0.400 | 0.300 | 0.200 | 0.100 |
| 96 | 0.097 | 0.941 | 0.939 | 0.184 | 0.688 | 0.603 | 0.519 | 0.435 | 0.350 | 0.266 | 0.181 |
| 144 | 0.187 | 0.896 | 0.899 | 0.341 | 0.684 | 0.613 | 0.542 | 0.471 | 0.400 | 0.329 | 0.258 |
| 192 | 0.271 | 0.855 | 0.864 | 0.476 | 0.680 | 0.621 | 0.563 | 0.504 | 0.446 | 0.388 | 0.329 |
| 240 | 0.348 | 0.813 | 0.831 | 0.588 | 0.673 | 0.627 | 0.580 | 0.534 | 0.487 | 0.441 | 0.394 |
| 288 | 0.418 | 0.767 | 0.798 | 0.677 | 0.662 | 0.628 | 0.593 | 0.558 | 0.523 | 0.488 | 0.453 |
| 336 | 0.481 | 0.718 | 0.766 | 0.744 | 0.647 | 0.623 | 0.599 | 0.576 | 0.552 | 0.528 | 0.504 |
| 384 | 0.540 | 0.672 | 0.737 | 0.803 | 0.632 | 0.619 | 0.606 | 0.593 | 0.580 | 0.566 | 0.553 |
| 432 | 0.601 | 0.636 | 0.717 | 0.868 | 0.625 | 0.622 | 0.618 | 0.615 | 0.611 | 0.608 | 0.604 |
| 480 | 0.660 | 0.603 | 0.699 | 0.931 | 0.620 | 0.626 | 0.632 | 0.637 | 0.643 | 0.649 | 0.655 |
| 528 | 0.716 | 0.568 | 0.681 | 0.983 | 0.612 | 0.627 | 0.642 | 0.657 | 0.671 | 0.686 | 0.701 |
| 576 | 0.758 | 0.517 | 0.656 | 0.999 | 0.589 | 0.614 | 0.638 | 0.662 | 0.686 | 0.710 | 0.734 |
| 624 | 0.778 | 0.435 | 0.620 | 0.958 | 0.538 | 0.573 | 0.607 | 0.641 | 0.675 | 0.710 | 0.744 |
| 672 | 0.792 | 0.346 | 0.585 | 0.908 | 0.480 | 0.524 | 0.569 | 0.614 | 0.658 | 0.703 | 0.748 |
| 720 | 0.815 | 0.274 | 0.559 | 0.885 | 0.436 | 0.490 | 0.544 | 0.598 | 0.652 | 0.706 | 0.761 |
| 768 | 0.851 | 0.231 | 0.545 | 0.900 | 0.417 | 0.479 | 0.541 | 0.603 | 0.665 | 0.727 | 0.789 |
| 816 | 0.909 | 0.227 | 0.544 | 0.965 | 0.431 | 0.500 | 0.568 | 0.636 | 0.704 | 0.772 | 0.841 |
| 864 | 0.954 | 0.203 | 0.536 | 1.000 | 0.428 | 0.503 | 0.578 | 0.653 | 0.728 | 0.803 | 0.878 |
| 912 | 0.957 | 0.113 | 0.510 | 0.943 | 0.367 | 0.451 | 0.535 | 0.620 | 0.704 | 0.789 | 0.873 |
| 960 | 0.959 | 0.022 | 0.485 | 0.888 | 0.303 | 0.397 | 0.491 | 0.584 | 0.678 | 0.772 | 0.866 |
| 1008 | 1.000 | 0.000 | 0.479 | 0.918 | 0.300 | 0.400 | 0.500 | 0.600 | 0.700 | 0.800 | 0.900 |

**Table 1.** Comparison of Fittingness (FN), Parallel Efficiency (Eff) and Energy-Delay Product (EDP) metrics across different processor counts for the Integrated Forecasting System (IFS) atmospheric component. The results illustrate how the recommended parallelisation to use for this component changes with the Time-To-Solution weight ($TTS_w$), ranging 0.3 to 0.9.

experience fluctuations due to background system load, hardware failures, and network congestion. The HPC platform used, MareNostrum4, was continuously monitored, and operations receive alerts when performance falls below expected levels. Any jobs executed during these periods can be identified and discarded to prevent skewed results. Additionally, and to further minimise the impact of these factors and ensure the reliability of our performance analysis, we have followed practices described below:

- Exclusive resource allocation: All jobs were submitted with the "*-exclusive*" clause, which ensures allocated nodes were not shared with other running jobs. This minimises performance noise from co-scheduled workloads.

- Simulation length: We configured model runs to use longer simulation chunks, which helps smooth out machine performance fluctuations. Depending on the model speed, we have chosen different chunk sizes. For SR runs, we used 1-year chunks, whereas for HR using 3-month chunks was enough. IN both cases each chunk has a runtime of ~1h.

- Multiple runs: Each resource configuration (chunk) was executed at least twice, and the results were averaged to mitigate fluctuations.

- Ignore the initialisation and finalisation phases: The initialization and finalization phases of an ESM run involve a higher proportion of I/O operations for reading initial conditions and writing outputs, making them less representative of sustained model performance. To account for this, we analysed the runtime deviation of these phases compared to the regular time-stepping loop and found that discarding the first and last simulated day was sufficient to account only for the regular timesteps. This was easily achieved using a dedicated parameter in the load balance analysis tool integrated in EC-Earth3 (Maisonnave et al., 2020).

- Run on different moments of the day: To account for diurnal fluctuations in HPC load, experiments were executed at different times. This was not strictly enforced but naturally resulted from using a queue that allowed only one job per user at a time. Combined with varying queue wait times, this led to experiment jobs running at different times throughout the day.

- Manual and post-mortem validation: All reported results underwent manual validation. Additionally, once an optimized resource setup was identified, a duplicate run was performed to confirm that the observed performance improvement was consistent with the initial measurement.

## 4   Automatic load-balance method

The following section describes the auto-lb, a methodology and tool aimed at improving the performance of ESMs by determining the proper distribution of the HPC resources (PEs) allocated to each component in coupled executions. This is accomplished by minimizing the core-hours lost due to synchronizations between interacting coupled components and by selecting a well-balanced speed for the coupled execution, considering the different scalability properties of the individual components. Additionally, given that different platforms and users may have varying constraints and criteria, the method can be used to find a solution within a restricted maximum number of PEs. It also allows users to define the priority between the achieved speed (TTS) and the core-hours consumed (ETS), as described in Section 3.2. The methodology is described in more detail in Algorithm 1 and Figure 3. It and can be divided into 3 main steps:

1. Get component's scalability: Obtain the SYPD (i.e. execution time) for each component involved in the coupled configuration using various PEs counts. The goal is to have a real representation of the model's performance. Thus, it is recommended to take the metrics from a configuration as similar as possible to the coupled run (same resolution, modules, IO, online diagnostics, compilation fags, etc.), and getting as close to the actual model performance. To minimise measurement uncertainties, we followed the best practices outlined in Section 3.3.

2. Prediction script: A Python script that, given the scalability curves of the components involved in the coupled configuration, returns the best resource allocation (i.e. how many PEs have to be assigned to each coupled component) depending on the user criteria (TTS or ETS) using the FN metric.

3. Load-balance workflow: Workflow that will submit multiple instances of the ESM on the HPC machine using an exist-
     ing climate workflow manager called Autosubmit workflow manager (AS) (Manubens-Gil et al., 2016). The workflow
     involves an iterative process, with each step involving the following: submitting multiple instances of the ESM, each
     with different resource configurations (initially, the resource allocations used is the one estimated as "optimal" by the
     Prediction scrip), collecting the performance metrics from each run, and making fine-grain modifications to the resource
setup to reduce the coupling cost (i.e., including the waiting time due to synchronisations and the time spent performing
     interpolations on the fields being exchanged) of the ESM at the next iteration. The performance achieved by each run is
     stored, and the outcome of the workflow is the resource setup which outperformed all the others. To mitigate performance
     uncertainties that could affect the results, we applied the practices described in Section 3.3.

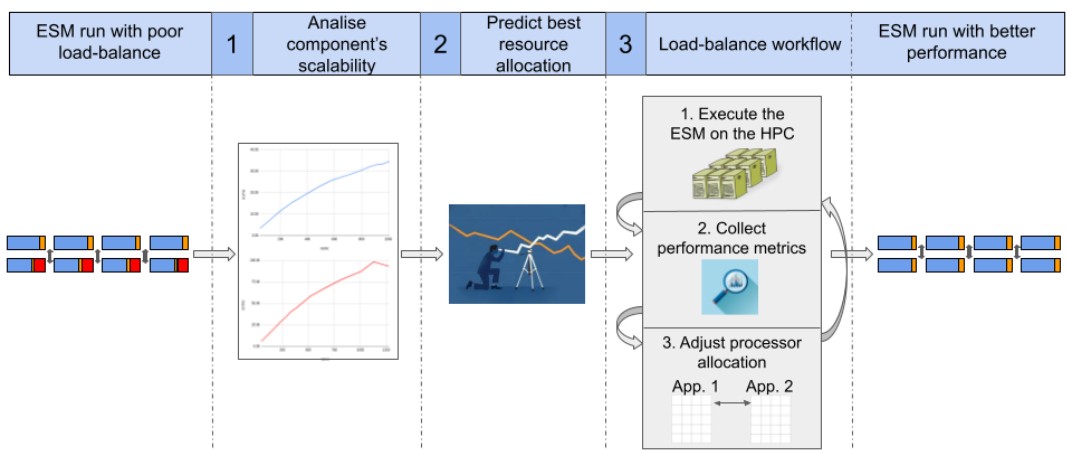

**Figure 3.** An overview of the auto-lb workflow, illustrating the steps to enhance the performance of an Earth System Model (ESM) from
an initially unbalanced configuration. The process begins with (1) obtaining the scalability properties of each component (scalability curve).
These results are then used by (2) the prediction script to estimate potential well-balanced resource configurations. Finally, these configura-
tions are used in (3) to iteratively simulate multiple instances of the ESM to identify a solution that minimises the coupling cost, potentially
improving the SYPD (i.e., speed) and reducing the CHSY (i.e., computing cost) of the simulation.

## 4.1 Prediction script

The number of possible configurations that can be used in coupled ESMs is too large to individually test each one. Take, for
     instance, 2-component systems like IFS-NEMO experiments, where both can utilize from 1 to 21 nodes. Using a granularity of
     1 node, there are $20 \times 20 = 400$ possible solutions. However, most of these configurations are completely unbalanced. Testing
     all of them is unnecessary and would result in a waste of resources with no added value.

     The Prediction script can search in this solution space in less than one second, approximating the results from each combi-
nation of PEs for IFS-NEMO based on the prior knowledge of the parallel behaviour that we have from the scalability curves.

Thus, finding the best setups for the TTS/ETS criteria selected. This not only ensures well-balanced setups but also considers potential high-quality parallelisation regions for individual components.

NEMO nprocs

| IFS nprocs | 48 | 96 | 144 | 192 | 240 | 288 | 336 | 384 | 432 | 480 | 528 | 576 |
|---|---|---|---|---|---|---|---|---|---|---|---|---|
| 48 | 0.5 | - | - | - | - | - | - | - | - | - | - | - |
| 96 | - | 0.54 | 0.45 | 0.36 | - | - | - | - | - | - | - | - |
| 144 | - | 0.6 | 0.59 | 0.52 | 0.46 | 0.4 | 0.33 | 0.27 | 0.21 | 0.14 | - | - |
| 192 | - | 0.53 | 0.69 | 0.64 | 0.59 | 0.54 | 0.49 | 0.44 | 0.39 | 0.34 | 0.29 | 0.24 |
| 240 | - | 0.46 | 0.7 | 0.72 | 0.68 | 0.64 | 0.6 | 0.56 | 0.52 | 0.48 | 0.43 | 0.39 |
| 288 | - | 0.39 | 0.66 | 0.8 | 0.76 | 0.72 | 0.69 | 0.65 | 0.62 | 0.58 | 0.55 | 0.51 |
| 336 | - | 0.32 | 0.61 | 0.81 | 0.81 | 0.78 | 0.75 | 0.71 | 0.68 | 0.65 | 0.62 | 0.59 |
| 384 | - | 0.25 | 0.57 | 0.77 | 0.81 | 0.78 | 0.75 | 0.72 | 0.69 | 0.66 | 0.63 | 0.6 |
| 432 | - | 0.19 | 0.52 | 0.74 | 0.83 | 0.8 | 0.77 | 0.74 | 0.71 | 0.68 | 0.65 | 0.63 |
| 480 | - | - | 0.48 | 0.71 | 0.87 | 0.87 | 0.85 | 0.82 | 0.8 | 0.77 | 0.74 | 0.72 |
| 528 | - | - | 0.43 | 0.67 | 0.84 | 0.9 | 0.88 | 0.85 | 0.83 | 0.8 | 0.78 | 0.75 |
| 576 | - | - | 0.39 | 0.64 | 0.82 | 0.85 | 0.82 | 0.8 | 0.77 | 0.75 | 0.72 | 0.69 |

**Table 2.** Fittingness matrix for IFS-NEMO coupled execution using a $TTS_w$ of 0.5. The matrix shows the Fittingness metric for various processes combinations, with IFS PEs in the vertical axis and NEMO PEs along the horizontal. Cells are color-coded from red (worst) to green (best), with gray indicating configurations worse than the baseline setup of 48 processes (equivalent to 1 node) per component.

Table 2 illustrates the Fittingness metric using a $TTS_w$ of 0.5. The best solution found uses 528 PEs for IFS and 288 for NEMO. However, these results are derived from scalability curves rather than actual simulations, which means they might not account for all factors that influence HPC machine performance. The Prediction script addresses this by providing not just the single best configuration, but the top N possible configurations (to be set by the user). This approach balances the risk of limited search space if only one combination is considered against the impracticality of exploring every possible configuration. Analysing the entire search space would be excessively time-consuming and computationally costly, potentially outweighing the gains of any performance analysis. For instance, and as detailed in Section 5, we have found that selecting the top 5 configurations provides a reasonable balance between total auto-lb runtime (around 24 hours) and search space given the application under study (EC-Earth). In the given example (Table 2), the top 5 combinations are 528-288, 528-336, 480-240, 480-288 and 480-336 (IFS-NEMO). Extreme combinations with no practical benefit are grayed out, as their huge coupling cost (vast difference in execution time between the coupled components), which makes them less performant than the baseline case (1 node per component). These configurations are not worth further investigation.

The Prediction script will, therefore, serve as a guide for the Load-balance workflow (Section 4.2) as now it won't have to search in the whole solution space but only in a relatively confined space to find the best resource configuration. This minimises the resource wastage by avoiding testing all possible combinations of PEs for IFS and NEMO and starting the in-depth analysis with some already potentially good setups.

## 4.2 Load-balance workflow

The Load-balance workflow is an iterative process that consists of a loop that submits multiple instances of the experiment using different resource setups, from which it collects the metrics defined in Section 3. These metrics guide the redistribution of the computational resources assigned to each component in subsequent iterations to minimise the coupling cost while improving the overall performance. This reallocation policy relies on the $partial\_coupling\_cost$ metric (Equation 5), which identifies the component contributing the most to the coupling cost. The identified component, referred to as the *donor*, is the

one underutilising its allocated resources while waiting for coupling data from the other component, labeled as *recipient*. The number of PEs transferred between the *donor* and the *recipient* at the first (*initial_step*) and last (*minimum_step*) iterations is a user-defined parameter, and depends on the application's sensibility to changes in the number of parallel resources. The outcome of the current iteration is a new set of resource configurations which will be submitted in the following iteration. The workflow is designed to guarantee convergence. If, at a given iteration, the direction of resource transfer changes (i.e.,

a previously identified recipient now becomes a donor), the step size is reduced by half. This iterative refinement continues until the step size falls below a user-specified threshold (*minimum_step*), at which point no further meaningful adjustments can be made. The workflow concludes when no new viable configurations can be explored or when further refinements produce negligible differences in performance. At this stage, the Fittingness (FN) metric is evaluated across all tested configurations to determine the best resource allocation found.

Fig. 4 provides an overview of a single workflow iteration that runs 2 instances per resource configuration.

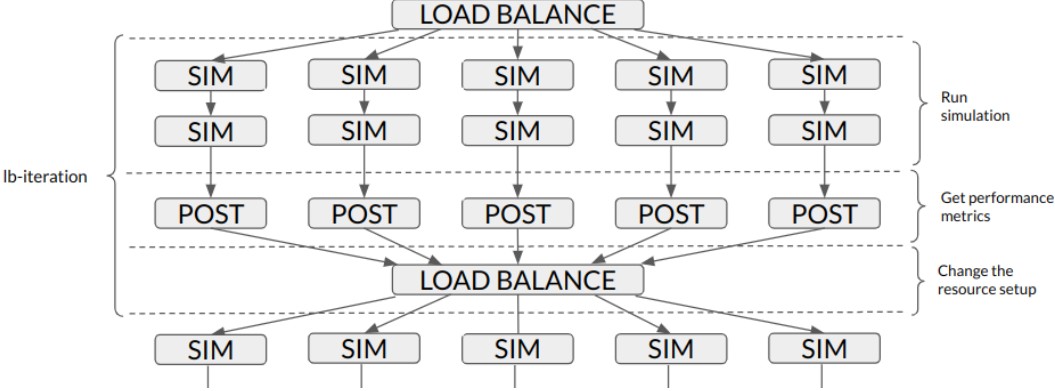

**Figure 4.** Overview of a single iteration of the load-balance workflow (lb-iteration). 5 different resource configurations (SIM) are submitted, running 2 instances for each. The performance results are gathered in the POST_LUCIA job and the LOAD_BALANCE job will give resources from one component to the other to achieve better balanced configuration.

---

**Algorithm 1** Automatic Load-Balance Method (auto-lb)

---

1: **Given:**
2:     A set of $T$ independent tests with different resource configurations: $tests\_to\_explore = \{test_0, test_1, \ldots, test_T\}$
3:     An initial step size: $initial\_step$
4:     The minimum allowable step size: $minmum\_step$
5:     A step size for each test, initially set to $initial\_step$: $\{step_0, step_1, \ldots, step_T\} = initial\_step$
6: **while** $tests\_to\_explore \neq \emptyset$ **do**                                                   ▷ Start lb_iter
7:     **for** each *test* in *tests_to_explore* **do**
8:         Submit the *test* to the HPC platform                                      ▷ SIM job
9:         Collect the performance metrics                                            ▷ POST job
10:     **end for**
11:     Identify the *donor* and *recipient* components                          ▷ Start LOAD_BALANCE job
12:     Define a new resource setup by reallocating $S$ PEs from the *donor* to the *recipient*
13:     Check that the new resource setup has not been tested before
14:     **if** the new setup was executed before on any of the previous tests **then**
15:         Halve the step size for this *test*: $step_i = step_i/2$
16:     **end if**
17:     **if** $step_i \geq minimum\_step$ **then**
18:         Submit test with the new configuration (jump to 7.)
19:     **else**
20:         Remove this *test* from *tests_to_explore*
21:     **end if**                                                                      ▷ End LOAD_BALANCE job
22: **end while**                                                                       ▷ End lb_iter

---

## 5    Results and discussion

In this section, we present the results of using the auto-lb tool to different configurations and experiments to demonstrate its effectiveness at improving ESMs performance and its versatility across different resolutions and platforms. We begin by evaluating the standard-resolution EC-Earth3 configurations used in the CMIP6 exercise, highlighting how our tool improves
upon configurations previously considered to be near-optimal. Next, we analyse high-resolution EC-Earth3 configurations used in the European Climate Prediction system (EUCP), showcasing the tool's capability to handle scenarios that require significantly larger computational resources. Additionally, we explore how varying the trade-offs between Time-To-Solution (TTS) and Energy-To-Solution (ETS) can yield different well-balanced configurations, depending on the specific needs of each experiment and the HPC platform. We also illustrate the portability of auto-lb by using it on different HPC platforms, such as
MN4 and CCA, demonstrating its adaptability in improving ESMs performance across diverse computational environments.

## 5.1 EC-Earth3 at standard resolution for CMIP6

During the CMIP6 project, even when accounting only for experiments used for production (not taking into account the spin-up runs), more than 240000 years were simulated for multiple ESM and across different HPC platforms. At the BSC, an EC-Earth3 SR CMIP6 configuration was used to execute more than 14000 years in MareNostrum4 (Acosta et al., 2023a). Thus, achieving the best performance was crucial for such a big project. An "optimal" resource configuration of 384 PEs for IFS and 240 for NEMO was agreed upon. This configuration resulted in a total number of PEs lower than 768. This value is significant because the scheduling policy permits jobs utilising up to 768 PEs to access a "debug" queue. While this reduces queue time, it restricts the scheduler to allow no more than one job to run simultaneously for each HPC user. The average performance results for one chunk with this configuration was 15.29 SYPD, 1113 CHSY and had a coupling cost of 14.81%. Figure 5a shows the scalability of IFS. The model scales well until 350 processes and seems to saturate at 550. Figure 5b demonstrates that NEMO scales exceptionally well. The cost of adding more parallel resources remains negligible until 600 PEs. Beyond this point, the speed-up gains become less pronounced compared to earlier increments.

After setting up the experiment and obtaining the scalability curves for IFS and NEMO, the Prediction script was executed with the following parameters: a max_nproc set to 672 ( the maximum PEs for IFS and NEMO after subtracting the 95 used by XIOS and 1 used by RNF, $672 + 95 + 1 = 768$), and a $TTS_w$ of 0.5. These parameters are explained in Section 4.1.

The Prediction script found the "optimal" to be 384 PEs for IFS and 264 for NEMO. The top 5 configurations are shown in Table 3. The result of the workflow is illustrated in Figure 6. Tests from 0 to 4 are resource configurations given by the prediction script and test 5 is the original one. The load-balance workflow finished after 4 iterations and a total of 24 (6x4) resource configurations have been tested. Note, however, that as shown in Figure 6, four of the tests are repeated (lb-iter 3, tests 0,3,4 and 5). The total execution time of the workflow has been 50 hours. The best result is 408 IFS - 240 NEMO, which compared to the original configuration is 4.7% faster (16.01/15.29) and 1.3% less costly (1099/1113). The coupling cost grows from 14.81% to 17.4% but it is compensated by using a better number of PEs given NEMO and IFS scalability properties. If the resource configuration found by the auto-lb had been used during the CMIP6 exercise, achieving a performance increase of 4.7% in execution time is equivalent to reducing the simulated time by $14020/15.29 - 14020/16.01 = \sim 41$days (if experiments were run by only one user). Similarly a reduction of the cost by 1.3% is equivalent to the cost of simulating 182 years.

The results also demonstrate the high accuracy of the Prediction script. As illustrated in the first row of Figure 6, which correspond to the resource configurations provided by the Prediction script (lb-iter 0), tests 0, 1, 2, and 3 consistently outperform the original setup (lb-iter 0, test 5). Therefore, the only predicted configuration performing worse than the original is observed in test 4. It is noteworthy that Figure 6 gives evidence that the iterative auto-lb phase leads to better resource setups. Following the evolution of test 4, after two lb-iterations (lb-iter 2, test 4) the auto-lb workflow achieved a new configuration which also outperforms the original one. Similarly, just 1 iteration after the original resource setup (lb-iter 1, test 5) shows that reallocating 48 processes from IFS to NEMO also provide a superior configuration compared to what was used in production during CMIP6.

|       | 1   | 2   | 3   | 4   | 5   | orig |
|-------|-----|-----|-----|-----|-----|------|
| IFS   | 384 | 360 | 408 | 408 | 408 | 384  |
| NEMO  | 264 | 240 | 264 | 240 | 216 | 240  |

**Table 3.** Top 5 initial resource configurations from the Prediction script to be used by the load-balance workflow for the SR CMIP6 experiment.

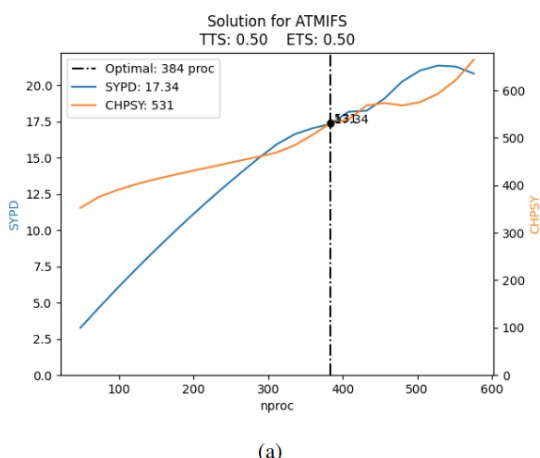

(a)

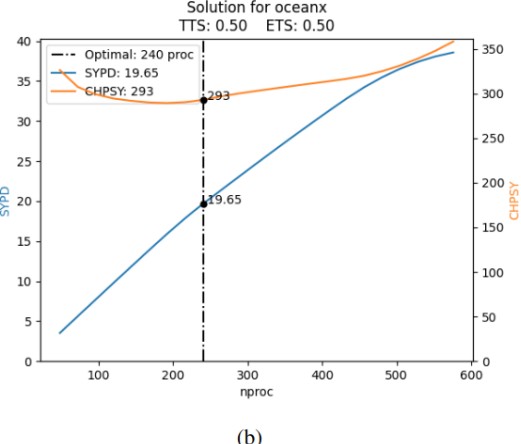

(b)

**Figure 5.** Scalability and predicted best resource allocation for IFS and NEMO at standard resolution for CMIP6 experiments

## 5.2 EC-Earth3 at high resolution for the European Climate Prediction system (EUCP)

During the European Climate Prediction system (EUCP) project, a high resolution experiment involving IFS and NEMO was used to simulate a total of 400 years. The configuration used for those experiments was 912 PEs for IFS and 1392 PEs for NEMO. Figure 7a shows the scalability of IFS. The CHSY does not increase much up to ∼500 processes. Almost achieving an ideal speed-up. After 500 processes, there seems to be some number of PEs better than others as the model SYPD curve is flat around 800, 900 and over 1000 PEs. Figure 7b shows the scalability of NEMO. We observe a superlinear speed-up as

the CHSY is reduced as the number of PEs increases. The component, however, has underperforms near 1000 PEs. And the execution cost starts increasing at the highest number of PEs configurations.

Table 4 shows the default and the top 5 resource configurations found by the Prediction script plus the test with the original resource setup used before the analysis (orig). The $max\_nproc$ allowed is 2400, the $TTS_w$ was set to 0.5. The load-balance workflow finishes after 5 iterations. The total execution time of the workflow is 50 hours (1 HourperTest * 2 TestsperConfigu-

430 ration * 5 InitialConfigurations * 5 lb-iterations = 50 hours).

Figure 8 shows the results of the auto-lb workflow. The performance of the original resource configuration, shown in lb-iter 0, test 5, was 3.54 SYPD, 16277 CHSY, and a coupling cost of 7.25%. The best solution is found in lb-iter 4, test 4, and

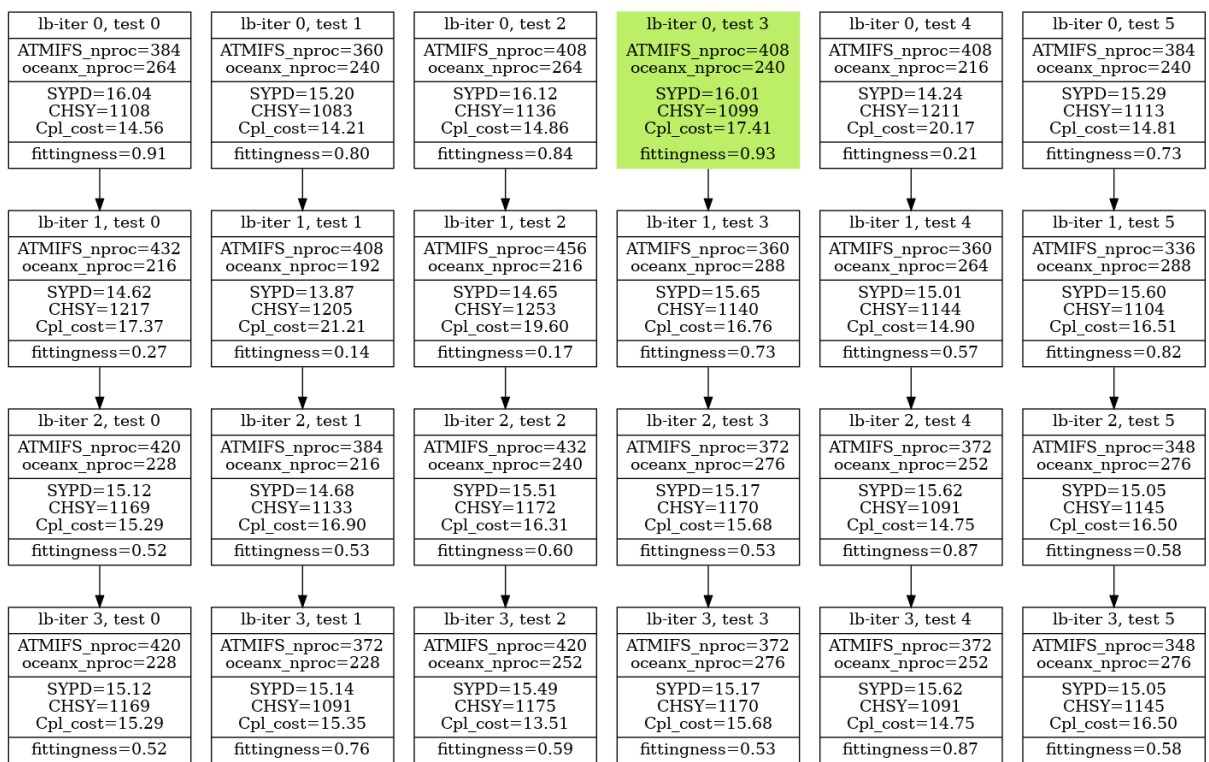

**Figure 6.** Performance results of each of the resource configurations tested to optimize a SR CMIP6 experiment. The metrics are the average of 3 runs of 6 months each.

|       | 1    | 2    | 3    | 4    | 5    | orig |
|-------|------|------|------|------|------|------|
| IFS   | 864  | 912  | 864  | 768  | 768  | 912  |
| NEMO  | 1389 | 1389 | 1437 | 1341 | 1389 | 1392 |

**Table 4.** Top 5 initial configuration from the Prediction script to be used by the load-balance workflow to find a better resource configuration for the EUPC HR experiment using a $TTS_w$ of 0.5.

achieves a performance of 3.48 SYPD and 15494 CHSY. This configuration is 1.7% slower than the original but reduces the execution cost by 4.9%. Moreover, note that there is also a new and better resource configuration found while trying to reduce the coupling cost for the original one, the lb-iter 3 test 5. This configuration uses 876 processes for IFS and 1428 for NEMO. The parallelization and the SYPD are the same as the original one but the CHSY is reduced by 363 (2.2%).

Having used this experiment to simulate the 400 years, reducing the CHSY by 4.9% is equivalent to save the executing cost of running 400*4.9% $\simeq$ 20 years (and more than 300,000 core-hours) with the same configuration.

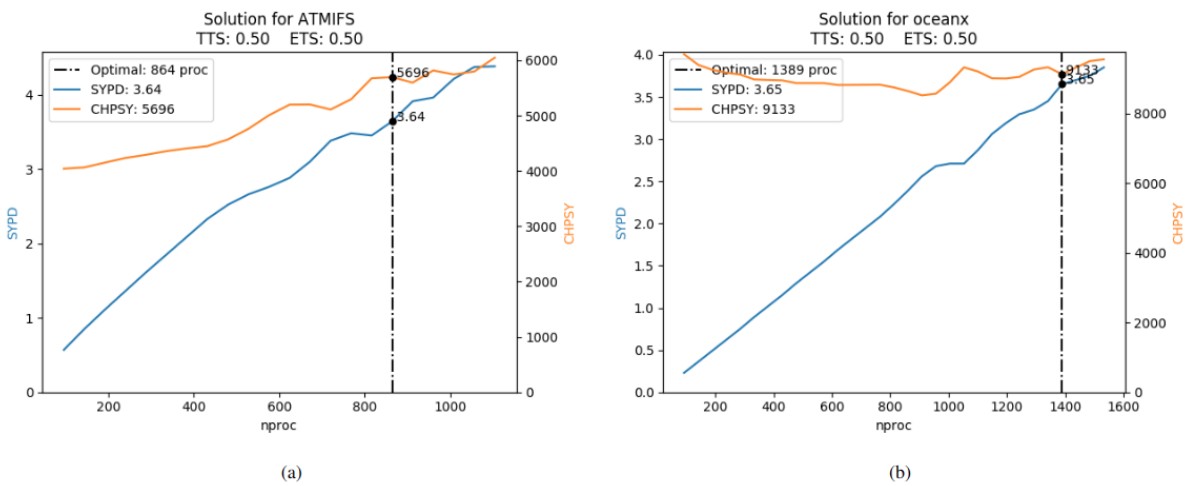

**Figure 7.** Scalability and predicted best PEs allocation for IFS and NEMO at high resolution for an EUCP experiment using a $TTS_w$ of 0.5

## 5.3 Time-to-Solution vs Energy-to-Solution

One of the novel features of auto-lb is the possibility to apply different performance-efficiency criteria depending on the context. This section demonstrates how using the $TTS_w$ parameter can affect the outcome of the same experiment configuration.

Using the auto-lb workflow with a default $TTS_w = 0.5$, we determined that the recommended setup for an EC-Earth3 experiment at standard resolution in the ECMWF machine (CCA HPC) is to use 684 PEs for IFS and 216 for NEMO. This configuration achieves 17.55 SYPD and consumes 1230 CHSY, with a coupling cost of 11.21%.

However, due to constraints on the number of core-hours allocated to the project on that machine, users require a more conservative setup that consumes fewer core-hours. This can be easily achieved by rerunning the auto-lb workflow using smaller $TTS_w$. For example, setting $TTS_w = 0.25$ ($ETS_w = 0.75$) provides a less costly configuration. Figure 9 presents the results of the workflow with a $TTS_w = 0.25$, starting from the following resource setups given by the Prediction script (IFS-NEMO): 576-144 468-108, 432-108, 468-144, 540-144 (see lb-iter 0).

The best configuration is found in lb-iter 3 test 2, utilising 423 PEs for IFS and 117 for NEMO. This configuration achieves 13.94 SYPD and 939 CHSY, with a coupling cost of 8.29 and used a total of 540 PEs. Compared to the solution found using $TTS_w = 0.5$, this setup reduces the speed of the ESM by 25.9% (17.55 vs. 13.94), but improves the CHSY by 31% (1230 vs. 939). Furthermore, the coupling cost is reduced by $11.21 - 8.29 = 2.92\%$ and fewer PEs are required to run. This can be visualised in Figure 10, which shows the recommended parallelisation for IFS (10a) and NEMO (10b) when changing the

$TTS_w$ from 0.5 to 0.25.

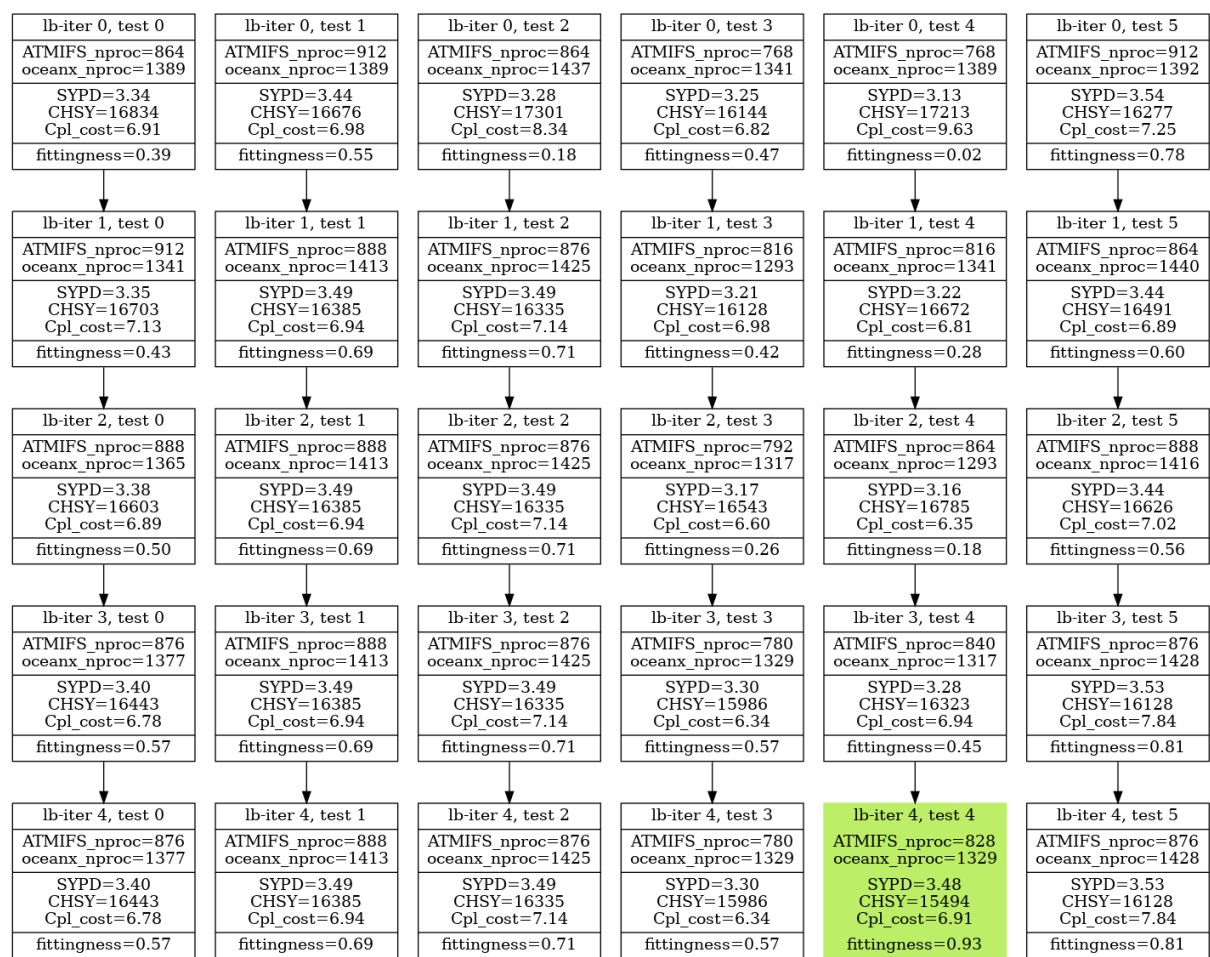

**Figure 8.** Performance results of each of the resource configurations tested to optimize a high-resolution EUCP experiment using a $TTS_w$ of 0.5. The metrics represent the average of 2 runs of 2 months each.

## 6   Conclusions

Coupled Earth System Models (ESMs) performance is limited by the load-balance between its constituents. While some works propose to deal with the problem by adapting the applications to support malleability, operational ESMs developed and maintained by different institutions in Europe mainly try to find the best resource configurations manually. Without an adequate methodology and an improved set of metrics for evaluating and addressing load imbalance, it has been demonstrated that coupled ESMs run with suboptimal resource configurations, leading to a diminishing of their speed and parallel efficiency.

This study introduces a novel methodology to improve resource allocation for each component in widely used EC-Earth3 experiments. The methodology includes a Prediction script to estimate the best possible solutions and an iterative process for running the simulations on the High Performance Computing (HPC) machine, collecting the performance metrics and making

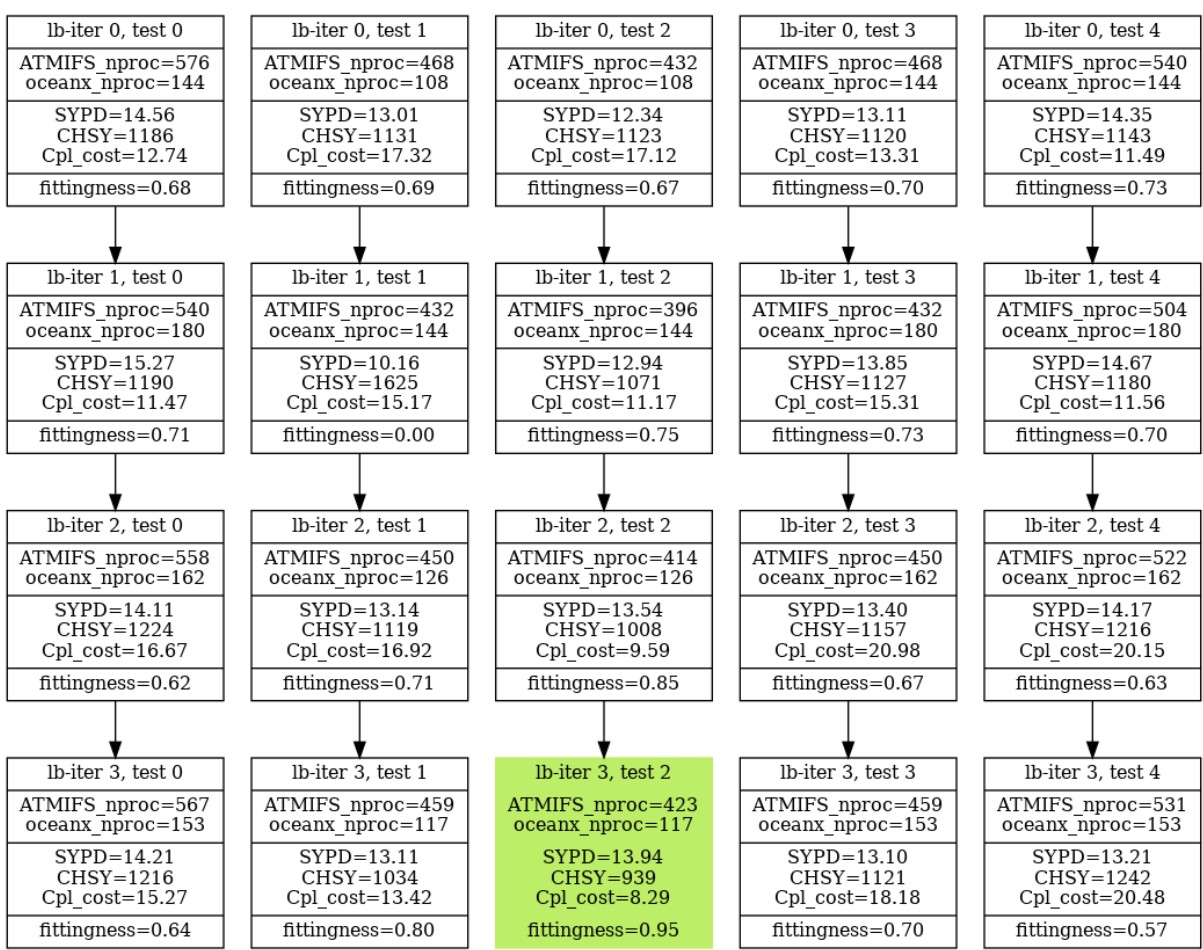

**Figure 9.** Performance results of each of the resource configurations tested to optimize a standard resolution experiment of EC-Earth3 using a $TTS_w$ of 0.25. The metrics are the average of 3 runs of 3 months.

fine-grain optimizations to mitigate the coupling cost. The methodology has been integrated into the Barcelona Supercomputing Centre (BSC) official workflow manager for EC-Earth3, Autosubmit workflow manager (AS), minimizing user intervention as much as possible. This integration allows any EC-Earth3 user using AS to easily take advantage of the auto-lb methodology on any of the other machines where the workflow manager is deployed (e.g., LUMI, MN5, MELUXINA, HPC2020, etc.). Additionally, auto-lb consists entirely of bash and Python scripts, making its core functionalities easily portable to other workflow managers or even runnable manually if required.

A new metric, named Fittingness, has been introduced to asses coupled ESMs performance. It allows to parameterise the energy/time trade-off. This flexibility enables the identification of multiple well-balanced solutions based on user-specific need, such as core-hours budged limitations, urgency in obtaining the output, etc.

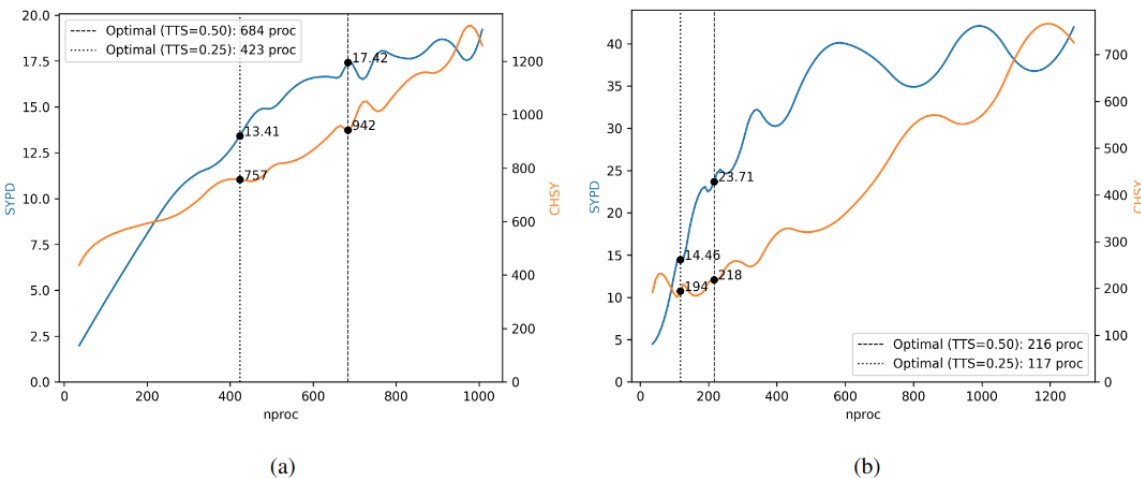

**Figure 10.** Scalability (SYPD and CHSY) and resource allocation for IFS (a) and NEMO (b) for an EC-Earth3 experiment at standard resolution on the ECMWF machine (CCA), using different Time-Energy criteria. For a $TTS_w = 0.50$, the coupled run should utilize 684 PEs for IFS and 216 for NEMO. For a $TTS_w = 0.25$, the coupled setup should be 423 PEs for IFS and 117 for NEMO.

The results demonstrate the portability of the auto-lb method across various HPC platforms, achieving improved resource configurations for different experiment configurations and resolutions. The authors believe that the best way to illustrate the usefulness of the proposed methodology is by showing its benefits for real and significant climate experiment that were carefully (manually) configured to maximize the performance. Therefore, Section 5.1 presents the computational improvements for a Coupled Model Intercomparison Project Phase 6 (CMIP6) experiment, which took months to simulate, covering over 14,000 years and consuming 15 million core-hours on MareNostrum4 (Acosta et al., 2023a). The results suggest savings of 4.7% of the execution time and a 1.3% reduction in core-hours needed. Similarly, Section 5.2 reports the results for high-resolution EC-Earth3 experiment used in the European Climate Prediction project (EUCP) project, simulating 400 years and consumed over 6.5 million core-hours on MareNostrum4. With the new resource setup achieved using the auto-lb methodology, the core-hours consumed could have been reduced by 4.9% at the expense of increasing the execution time by 1.7%. Alternatively, the method also provides another resource setup that maintains constant execution time but reduces the core-hours required by 2.2%. Finally, Section 5.3 presents two possible resource setups for EC-Earth3 on another HPC machine, European Centre for Medium-Range Weather Forecasts (ECMWF)'s CCA HPC. The two setups differ in the criteria used. For a more energy-efficient solution, the auto-lb methodology was used with a $TTS_w = 0.25(ETS_w = 0.75)$. This solution is 25.9% slower than using the default value of $TTS_w = 0.50$, but it reduces core-hours by 31% and uses fewer Processing Elements (PEs), demonstrating that both solutions are viable and allowing the user to choose the most appropriate one depending on the specific context in which it will run.

Looking ahead, it is expected that ESMs will continue to grow in complexity, incorporating more components to simulate more features of the Earth system. For instance, some EC-Earth3 configurations already couple up to 5 different components

simultaneously, resulting in a better representation of some Earth phenomena but increasing the load-imbalance significantly. Big upcoming international projects like the Coupled Model Intercomparison Project Phase 7 (CMIP7) are crucial for the

495 advance of climate science, but they come at the expense of a significant power consumption for computing. As shown, even with two-component systems, the solution space can easily grow into the multiple hundreds of different resource setups. Adding more components exponentially increases this solution space, making the usage of manual tuning and traditional methods even more limited with future complex simulations. This underscores the necessity for developing of more sophisticated tools like auto-lb.

At the same time, the increasing adoption of GPU acceleration in Earth System modelling software reflects a broader shift towards hybrid computing infrastructures. A good example of this trend can be found in the new EuroHPC systems, where seven out of the eight integrate both CPU and GPU resources. Consequently, methodologies for load-balancing must evolve to account for these new hybrid architectures. While the principles described for the auto-lb approach remain relevant, heterogeneous CPU-GPU codes introduce additional complexities. The primary challenge lies in controlling the speed at which

each component must run to maintain load balance. In a pure MPI setup, resource redistribution is straightforward, as coupled components share a common pool of processing elements (PEs, physical cores) and can reallocate them while keeping the total amount of parallel resources used constant. In contrast, for components running on different hardware (e.g., CPUs and GPUs), the term "processing element" has different meanings, and resources are not directly interchangeable—a CPU core and a GPU core do not have a one-to-one equivalence. Extending the auto-lb methodology to hybrid CPU-GPU ESMs would require a

standardised definition of computational resources. Such a definition could enable the optimisation process to account for the equivalences and differences between CPUs and GPUs, potentially through an application-specific equivalent compute unit metric. This metric would involve profiling the performance characteristics of each component on both types of hardware to guide resource allocation decisions.

Moreover, it is important to highlight that some climate models include additional parallelization parameters that influence

performance but have not been explicitly addressed in this manuscript. These include the ability to define different processor layouts (e.g., for 32 PEs, possible configurations could be 1×32, 2×16, 4×8, etc.) and the use of hybrid MPI-OpenMP parallelism. At present, these aspects are not managed within the auto-lb tool as we expect that this is already known before the balancing. It is important to emphasize that our methodology specifically addresses the load-balancing issues rather than optimizing the standalone performance of individual components. In cases where processor layouts or hybrid configurations

must be considered, we first conduct an exploratory standalone performance analysis of each component to determine the most efficient processor layout and the optimal number of OpenMP threads per MPI rank. These parameters are then treated as fixed throughout the load-balancing process. If these cases become more readily available, it is possible to update the workflow for better handling.

To ensure efficient use of HPC resources, auto-lb functionalities must be extended to support increasingly complex coupled

configurations and models running on hybrid computing infrastructures. The combination of heuristics through a prediction script with the automatic iterative process (running the ESMs and collecting novel performance metrics) offer an efficient approach to find better resource setups for coupled ESMs, while minimising the time and core-hours needed to find them.

*Code availability.* The source code for the prediction script is publicly available at: https://doi.org/10.5281/zenodo.14163512 (Palomas, 2024).

The EC-Earth3 source code is accessible to members of the consortium through the EC-Earth development portal. Access to the EC-Earth3 source code can be requested from the EC-Earth community via the EC-Earth website: http://www.ec-earth.org (last access: 18 November 2024). Model codes developed at ECMWF, such as the IFS atmospheric model, are the intellectual property of ECMWF and its member states. Therefore, access to the EC-Earth3 source code requires signing a software license agreement with ECMWF. The version of EC-Earth used in this study is tagged as 3.3.3.1 in the repository.

The Autosubmit workflow manager is available as a Python package on PyPi (https://pypi.org/project/autosubmit/, last access: 18 November 2024), with its documentation and user guide hosted at https://autosubmit.readthedocs.io/en/master/ (last access: 18 November 2024).

*Data availability.* No datasets were generated or analysed during the current study, so there are no data available for publication.

*Author contributions.* Sergi Palomas was the primary developer of the tool and methodology presented in this manuscript. Mario C. Acosta served as the principal investigator leading this research. Etienne Tourigny provided support and advice for setting up and running the experiments shown in the results section. Gladys Utrera contributed to the writing of the manuscript.

*Competing interests.* The authors have no competing interests to declare.

*Acknowledgements.* This research was supported by the European High Performance Computing Joint Undertaking (EuroHPC JU) and the European Union (EU) through the ESiWACE3 project under grant agreement No. 101093054, as well as the ISENES3 project under grant agreement No. H2020-GA-824084. Additional funding was provided by the National Research Agency through the OEMES project (PID2020-116324RA-I00).

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
