# Peer review of "Reducing Time and Computing Costs in EC-Earth: An Automatic Load-Balancing Approach for Coupled ESMs"

_Geoscientific Model Development, 2024_

## Referee Comment (RC3)

This article presents an automatic load-balancing approach for coupled ESM runs and evaluates it based on EC-EARTH. The methodology can be easily understood.

I think the following points should be addressed in the next revision.

1. Although only EC-EARTH is used in the evaluation, the approach proposed should be somehow common to other ESMs. So related works (such as the following list) should be referenced and compared, to show why the approach in this article is novel, more advanced or more effective.

1) D. Kim, J. W. Larson, and K. Chiu, "Automatic performance prediction for load-balancing coupled models," in Cluster, Cloud and Grid Computing (CCGrid), 2013 13th IEEE/ACM International Symposium on. IEEE, 2013, pp. 410–417.
2) CESMTuner: An Auto-Tuning Framework for the Community Earth System Model
3) An automatic performance model-based scheduling tool for coupled climate system models
4) Machine-Learning-Based Load Balancing for Community Ice Code Component in CESM
5) https://esmci.github.io/cime/versions/cesm2.2/html/misc_tools/load-balancing-tool.html

2. When obtaining the SYPD (i.e. execution time) for each component model under different parallel settings, how long of the simulation should be, and should the initialization cost be neglected?

3. There should be some conditions for using the new approach especially for prediction script. For example, the model run should be stable enough, which means multiple runs of the same model setting get adjacent runtime. If a HPC runs many applications at the same time, runs of the same model setting may not be stable enough.

4. Does NEMO or IFS have different processor layouts under the same number of PEs. For example, many models have been parallelized based on the decomposition on both X and Y directions. Given 32 PEs, the processor layout can be 1x32, 2x16, 4x8, ⋯ Moreover, some models may use hybrid MPI and OpenMP. Should the resource configuration take consideration of such kind of processor layout?

5. What is the key idea of the load-balance workflow? Is there any risk of failure in convergence? If there is no risk, why?

6. The word "optimal" has been used many times in the context. How to prove the "optimal" configuration is absolutely optimal? "near optimal" may be better.

7. Some words in Figure 3 should be improved. For example, "ESM with poor load-balance" should be "ESM run with poor load-balance".

8. Couplers like OASIS-MCT support flexible coupling lags which can transform concurrent run among component models into sequential run where one component model will wait in the whole run of another component model. Can the approach in this article be effective for such kind of ESM run?

---

## Author Comment (AC4)

**Reducing Time and Computing Costs in EC-Earth:**
**An Automatic Load-Balancing Approach for Coupled ESMs**

**by**

Sergi Palomas, Mario C. Acosta, Gladys Utrera, and Etienne Tourigny

**Reviewer (R)**
Authors (A)
* * *
R:
The paper addresses a very significant issue of load imbalances on (large) parallel runs of coupled ESM models. The proposed solution and the developed tool represent a meaningful contribution to alleviating the waste of computational resources and giving the users of the ESM models a better control over the coupled simulation runs and their overhead.

The current version of the paper is quite polished and, with an exception of a handful of mostly technical errors, nearly ready for publication.

A:
We sincerely appreciate the reviewer's insightful comments, which have helped improve the quality of our manuscript. Below, we provide detailed responses to each point raised.

**General comments**

R:
**A suggestion that the authors might want to briefly \*\*discuss in the outlook:\*\* How difficult would be an extension of the proposed methodology to heterogeneous/hybrid architectures (e.g. CPU-GPU systems)?**

A:
We agree that this should be discussed in the manuscript. We have added it in the outlook (lines

422-437). Now it reads:

"""

At the same time, the increasing adoption of GPU acceleration in Earth System modelling software reflects a broader shift towards hybrid computing infrastructures. A good example of this trend can be found in the new EuroHPC systems, where 7 out of the 8 integrate both CPU and GPU resources. Consequently, methodologies for load-balancing must evolve to account for these new hybrid architectures.
While the principles described for the auto-lb approach remain relevant, heterogeneous CPU-GPU codes introduce additional complexities.
The primary challenge lies in controlling the speed at which each component has to run to keep the load-balance. In a pure MPI setup, resource redistribution is straightforward, as coupled components share a common pool of processing elements (PEs, physical cores) and can reallocate them while keeping the total amount of parallel resources used constant.
In contrast, for components running on different hardware (e.g., CPUs and GPUs), the term "processing element" has different meanings, and resources are not directly interchangeable -a CPU core and a GPU core do not have a one-to-one equivalence.
The authors believe that the overall methodology described for auto-lb could be extended to hybrid CPU-GPU ESMs, provided that a standardised definition of the computational resources is established. This would allow the optimisation process to account for the equivalences and differences between CPUs and GPUs, potentially through an application-specific equivalent compute unit metric. Such a metric would involve profiling the performance characteristics of each component on both types of hardware to guide resource allocation decisions.
"""

**Specific comments:**

R:
**I suggest to include a column with parallel efficiency in the Table 1.**
A:
We have updated Table 1 to include the requested column for parallel efficiency. Additionally, we have added some missing information to the Table's caption.

**Technical comments:**

**R:**
**'can not' should be 'cannot' at several places in text**
A:
This has been corrected in all instances within the manuscript. (lines 152 and 176)

**R:**

**lines 178-181: since a single node is taken as the baseline, processors and processes should be replaced with nodes**

A:

We believe that comparing against a single node or the processors available within that node is conceptually equivalent. Since the granularity used in our tables, figures, and results is based on processors, we prefer to maintain the original terminology for consistency.

**R:**

**line 201: remove the multiplication dot in the denominator**

A:

This has been corrected in the manuscript.

**R:**

**lines 313-314: it seems that Figures 5a and 5b should be referenced there instead of Figures 10a and 10b**

A:

We confirm that there was an error in the references. This has been corrected in the appropriate section 5.1 (lines 313-314).

**R:**

**line 332: 'worse the original...' should read 'worse than the original...'**

A:

This has been fixed in the manuscript

**R:**

**Section 5.3: Figure 10 should probably be referenced somewhere within this section**

A:

As noted in a previous comment, there were inconsistencies in figure references in section 5.3. We have reviewed and corrected these references accordingly, as well as some minor improvements to the whole paragraph.

---

## Author Comment (AC5)

**Reducing Time and Computing Costs in EC-Earth:**
**An Automatic Load-Balancing Approach for Coupled ESMs**
**by**

Sergi Palomas, Mario C. Acosta, Gladys Utrera, and Etienne Tourigny

**Reviewer (R)**
Authors (A)
* * *
R:
This article presents an automatic load-balancing approach for coupled ESM runs and evaluates it based on EC-EARTH. The methodology can be easily understood.

A:
We sincerely appreciate the reviewer's insightful comments, which have contributed to improving the quality of our manuscript. In particular, we are grateful for the suggestions regarding the state of the art section, the description of our proposed approach, its applicability to previously unconsidered scenarios, and the technical corrections. Below, we provide detailed responses to each point raised.

**Comments**

R1:
**Although only EC-EARTH is used in the evaluation, the approach proposed should be somehow common to other ESMs. So related works (such as the following list) should be referenced and compared, to show why the approach in this article is novel, more advanced or more effective.**
1. D. Kim, J. W. Larson, and K. Chiu, "Automatic performance prediction for load-balancing coupled models," in Cluster, Cloud and Grid Computing (CCGrid), 2013 13th IEEE/ACM International Symposium on. IEEE, 2013, pp. 410–417.

We have improved the section citing the work by Kim et al. and explaining the differences with our work. The main difference is that their solution is for dynamic load-balancing approaches (solving the load-balance during runtime), which require malleability, and that they achieved that by extending the Model Coupling Toolkit (MCT). Thus, their solution is not available for most of the climate codes. Furthermore, the approach they developed is for an "integrated coupling framework", where components are subroutines merged into a single binary. Instead, what we propose is for coupling schemes where different binaries use a coupling library as a Multi-Program Multi-Data (MPMD) application. Finally, opposed to our work, their proposed method has not been validated against any state-of-the-art ESM, but rather to a "toy" model.

Specifically, now it reads (line 65):

```
Possibly the most notable contributions to dynamic approaches have been
done by Kim et al. 2011 to extend the Model Coupled Toolkit (MCT) to
Malleable MCT (MMCT), enabling malleability and incorporating a
load-balance manager module. This module decomposes the time of each
component during a Coupled Interval (CI) into constituent computation
and constituent coupling. The load-balance manager reallocates PEs from
the fastest (donor) to the slowest (recipient) component until solution
improvement ceases.
This work was further enhanced in Kim et al. (2012b), where MMCT was
extended with a prediction mechanism that maintains a database of
PEs-execution times at each iteration, and a manually-generated
heuristic optimisation to determine new resource configurations that
reduce the coupling step execution time. Kim et al. (2012a) extended
this approach to handle applications which have varying workloads during
the execution.

However, the manually-generated heuristic used for the prediction -based
on static and manual inspection of coupled model interaction patterns
and constituents' computations- becomes impractical for more complex,
realistic coupled models.
To address this, Kim et al. (2013) proposed an instrumentation-based
approach that collects runtime data from the constituents, demonstrating
how this information can be used to improve coupling performance and
accelerate the load-balancing decision-making process.

While these approaches have demonstrated significant improvements, they
are designed for a highly flexible coupling scheme applicable only to
climate models that adopt the MMCT extension of MCT. As a result, they
are not suitable for most state-of-the-art ESMs. Moreover, the method
proposed has not been validated with production ESMs used in climate
research, but rather with a simplified "toy" model that mimics a
simulation of the CESM.

In contrast, our proposed solution is not integrated into any specific
coupler, making it readily accessible to most ESMs used by the climate
research community that employs an external coupling library to link
multiple binaries (MPMD) into a single application.
```

2. (Nan et al. 2014) CESMTuner: An Auto-Tuning Framework for the Community Earth System Model

Added the following in the manuscript (line 83):

> *Static* load-balancing solutions are well suited for the climate science community due to the difficulties found in effectively applying dynamic approaches.
> One of the most significant contributions of *static* load-balancing is the work by Nan et al. (2019, 2014) for CESM, which introduced an auto-tuning component integrated into the CESM framework to optimise process layout and reduce model runtime. It achieves this by employing a depth-first search (DFS) method with a branch-and-bound algorithm to solve a Mixed Integer Nonlinear Programming (MINLP) problem, combined with a performance model of the model components to minimise search overhead. This approach improves upon the earlier method described by Alexeev et al. (2014), which relied on a heuristic branch-and-bound algorithm and a less accurate performance model.

3. (Nan et al. 2019) An automatic performance model-based scheduling tool for coupled climate system models

This work is from the same authors as the previous reference (Nan et al. 2014). We added the reference, and it has been included in the answer to R1.2

4. Balaprakash et al. (2015) Machine-Learning-Based Load Balancing for Community Ice Code Component in CESM

We have added the following to the manuscript (line 83, just after what has been added for R1.2 and R1.3):

```
Later, Balaprakash et al. (2015) proposed a static,
machine-learning-based load-balancing approach to find high-quality
parameter configurations for load balancing the ice component (CICE) of
CESM. The method involves fitting a surrogate model to a limited set of
load-balancing configurations and their corresponding runtimes. This
model is then used to efficiently explore the parameter space and
identify high-quality configurations. Their approach had to take into
account the six key parameters that influence CICE component
performance: the maximum number of CICE blocks and the block sizes in
the first and second horizontal dimensions (x, y); two categorical
parameters that define the decomposition strategy; and one binary
parameter that determines whether the code runs with or without a halo.
```

> They demonstrated that their approach required 6x fewer evaluations to
> identify optimal load-balancing configurations compared to traditional
> expert-driven methods for exploring feasible parameter configurations.

5. (Alexeev et al. 2014)
   https://esmci.github.io/cime/versions/maint-5.6/html/misc_tools/load-balancing-tool.html

   This load-balancing tool is based on the Alexeev et al. (2014) paper. Please see answer
   to R1.2 and R1.3

A:

Furthermore, we added these paragraphs after discussing the various previous works on CESM
(R1.2, R1.3, R1.4 and R1.5) to highlight how its coupling strategy differs from that of EC-Earth,
which uses an external coupling library (line 83, after all the paragraphs mentioned above):

> Importantly, coupling in CESM follows an *integrated coupling framework*
> strategy (Mechoso et al., 2021), where the climate system is divided into
> component models that function as subroutines within a single executable and
> orchestrated by a coupler main program (CPL7), which coordinates the
> interaction and time evolution of the component models. The coupler also
> allows for flexible execution layouts, enabling components to run
> sequentially, concurrently, or in a hybrid sequential/concurrent mode.
> This coupling strategy differs with approaches that use an external coupler
> (such as OASIS, MCT, or YAC), where existing model codes are minimally
> modified to interface with the coupling library and executed as separate
> binaries on different physical cores, either interleaved or concurrently.
> Furthermore, the performance model used requires generating and analysing
> execution traces to characterize the computation and communication patterns of
> key kernels for each coupled component separately. While this can provide more
> accurate performance representations, it also introduces significant
> challenges in adapting the approach to new components or other ESMs.

Moreover, we have also improved the paragraph that already appeared on static approaches
(line 84):

> Other *static* solutions, such as those proposed by Will et al. (2017) for the
> COSMO-CLM regional climate model and Dennis et al. (2012) for CESM,
> demonstrate that load balancing in widely used ESMs can be approached in a
> relatively simple manner. These methods aim to identify a resource
> configuration where all individual components run at roughly the same speed,
> often constrained by a predefined parallel efficiency threshold. However, as
> we will show, this approach can easily lead to suboptimal solutions.

And improved the sections where we presented our approach, which now better captures the difference with other related approaches (line 89):

> In this work, we present a *static* load-balancing method, the automatic load-balance (auto-lb), designed to improve resource allocation in coupled ESMs. Our approach is suited for coupled models that do not support malleability, where each component runs on separate cores as an MPMD application. Unlike other methods, our approach completely eliminates the need to modify any of the component's source codes; instead, it achieves load-balance by adjusting the allocation of PEs assigned to each component. To accomplish this, we have introduce two new performance metrics: firstly, the Partial Coupling Cost to quantify the cost of the coupling per component, and secondly, the Fittingness metric to better address the Energy-To-Solution (ETS, i.e. minimise the energy consumption) and Time-To-Solution (TTS, i.e. minimise the execution time) trade-off prevalent in all non-perfectly scalable applications (Abdulsalam et al., 2015).
> These advancements set our method apart from existing approaches that either focus exclusively on minimizing execution time (pure TTS) or enforce parallel efficiency thresholds that limit speed in an arbitrary manner.
>
> Moreover, the method includes a prediction phase capable of accurately estimating coupling performance based solely on the scalability properties of the individual model components. Unlike prior work, this eliminates the need for instrumenting the code, using profiling software, or trace generation. The results of the prediction phase allow our approach to significantly reduce the number of real simulations (and thus computational resources) required to determine an improved load-balancing configuration.
>
> Finally, the method is fully integrated in a workflow manager, ensuring that the process of identifying the best resource configuration requires minimal user intervention and aligns with standard practices in climate modelling.

R2:

**When obtaining the SYPD (i.e. execution time) for each component model under different parallel settings, how long of the simulation should be, and should the initialization cost be neglected?**

A:

Please, see answer for R3

**R3:**

**There should be some conditions for using the new approach especially for prediction script. For example, the model run should be stable enough, which means multiple runs of the same model setting get adjacent runtime. If a HPC runs many applications at the same time, runs of the same model setting may not be stable enough.**

A:

We share these concerns about model stability and the potential variability in a shared HPC

environment, as they are crucial for the reliability of both, the collected measurements and the results discussed in the manuscript.

To address these issues, we have already implemented several measures to ensure the precision of our performance metrics collection, which were not included in our original submission. In response to the reviewers' feedback, we have dedicated a new section to explicitly discuss these topics: *3.3 Model performance stability and measurement uncertainty*. We have added this section to the manuscript, and for reference, we include the full text below (line 232, after section *3.2 Time-to-Solution vs Energy-to-Solution criteria*):
* * *
**3.3 Model performance stability and measurement uncertainty**

Evaluating the performance of ESMs inherently involves uncertainty due to the variability of HPC environments. Under an ideal scenario, repeated runs of the same model setup should yield identical runtimes. However, in practice, HPC systems experience fluctuations due to background system load, hardware failures, and network congestion. The HPC platform used, MareNostrum4, was continuously monitored, and operations receive alerts when performance falls below expected levels. Any jobs executed during these periods can be identified and discarded to prevent skewed results. Additionally, and to further minimise the impact of these factors and ensure the reliability of our performance analysis, we have followed practices described below:

- Exclusive resource allocation: All jobs were submitted with the "*-exclusive*" clause, which ensures allocated nodes were not shared with other running jobs. This minimises performance noise from co-scheduled workloads.

- Simulation length: We configured model runs to use longer simulation chunks, which helps smooth out machine performance fluctuations. Depending on the model speed, we have chosen different chunk sizes. For SR runs, we used 1-year chunks, whereas for HR using 3-month chunks was enough. IN both cases each chunk has a runtime of ~1h.

- Multiple runs: Each resource configuration (chunk) was executed at least twice, and the results were averaged to mitigate fluctuations.

- Ignore the initialisation and finalisation phases: The initialization and finalization phases of an ESM run involve a higher proportion of I/O operations for reading initial conditions and writing outputs, making them less representative of sustained model performance. To account for this, we analysed the runtime deviation of these phases compared to the regular time-stepping loop and found that discarding the first and last simulated day was sufficient to account only for the regular timesteps. This was easily achieved using a dedicated parameter in the load balance analysis tool integrated in EC-Earth3 (Maisonnave et al., 2020).

```
    -  Run on different times of the day: To account for diurnal fluctuations
       in HPC load, experiments were executed at different times. This was not
       strictly enforced but naturally resulted from using a queue that allowed
       only one job per user at a time. Combined with varying queue wait times,
       this led to experiment jobs running at different times throughout the
       day.

    -  Manual and post-mortem validation: All reported results underwent manual
       validation. Additionally, once an optimized resource setup was
       identified, a duplicate run was performed to confirm that the observed
       performance improvement was consistent with the initial measurement.
```

**R4:**
**Does NEMO or IFS have different processor layouts under the same number of PEs. For example, many models have been parallelized based on the decomposition on both X and Y directions. Given 32 PEs, the processor layout can be 1x32, 2x16, 4x8, … Moreover, some models may use hybrid MPI and OpenMP. Should the resource configuration take consideration of such kind of processor layout?**

A:
IFS and NEMO don't offer the possibility of defining different processor layouts under the same number of PEs. However, this is available in other EC-Earth3 components and ESMs.
When processor layout variations are considered, the solution space expands significantly, increasing the complexity of exploring and finding the best combinations.
Under these scenarios, we first perform a standalone performance analysis of each component that allows different processor layouts. This preliminary analysis helps to identify an optimal configuration, which can guide whether specific parameters (e.g., keeping one dimension constant) or ratios should be maintained during the coupled analysis. Once determined, this information is used inside the load-balancing algorithm for special handling when redistributing PEs –particularly when the step (number of PEs to be reassigned between components) is not a multiple of the current processor layout.

A similar principle applies to hybrid MPI-OpenMP configurations. The ratio of OpenMP threads per MPI rank is generally established during the initial standalone performance analysis of each component and remains fixed throughout the load-balancing process.

It is important to emphasize that our methodology specifically addresses load-balancing in the coupled system rather than optimizing the standalone performance of individual components. While standalone component efficiency directly influences coupled performance, its detailed tuning is beyond the primary scope of our approach. We opted to not include this information as it does not apply to the configurations discussed in the manuscript. Nonetheless, we have

added some explanations to this in the *Conclusions* section. Which now reads (line 414)*:

> Moreover, it is important to highlight that some climate models include
> additional parallelization parameters that influence performance but have not
> been explicitly addressed in this manuscript. These include the ability to
> define different processor layouts (e.g., for 32 PEs, possible configurations
> could be 1×32, 2×16, 4×8, etc.) and the use of hybrid MPI-OpenMP parallelism.
> At present, these aspects are not managed within the auto-lb tool.
> It is important to emphasize that our methodology specifically addresses the
> load-balancing issues rather than optimizing the standalone performance of
> individual components. In cases where processor layouts or hybrid
> configurations must be considered, we first conduct an exploratory standalone
> performance analysis of each component to determine the most efficient
> processor layout and the optimal number of OpenMP threads per MPI rank. These
> parameters are then treated as fixed throughout the load-balancing process. If
> these cases become more readily available, it is possible to update the
> workflow for better handling.

* The *Counclusions* section has been significantly improved since the original version, incorporating feedback from Reviewer 1 as well. This paragraph appears right after a newly added one on how the proposed method would apply to hybrid CPU-GPU computation. For further details, please refer to the discussion with Reviewer 1.

**R5:**
**What is the key idea of the load-balance workflow? Is there any risk of failure in convergence? If there is no risk, why?**
A:
We have extended Section "*4.2 Load-balance workflow*" to explain better how the iterative process will always converge. Specifically, in line 290:

> The outcome of the current iteration is a new set of resource configurations
> which will be submitted in the following iteration.
> The workflow is designed to guarantee convergence. If, at a given iteration,
> the direction of resource transfer changes (i.e., a previously identified
> recipient now becomes a donor), the step size is reduced by half. This
> iterative refinement continues until the step size falls below a
> user-specified threshold ($minimum\_step$), at which point no further
> meaningful adjustments can be made. The workflow concludes when no new viable
> configurations can be explored or when further refinements produce negligible
> differences in performance. At this stage, the Fittingness (FN) metric is
> evaluated across all tested configurations to determine the optimal resource
> allocation.

**R6:**

**The word "optimal" has been used many times in the context. How to prove the "optimal" configuration is absolutely optimal? "near optimal" may be better.**

A:

We completely agree that the use of "optimal" was misleading. This has been fixed in all occurrences of the manuscript.

**R7:**

**Some words in Figure 3 should be improved. For example, "ESM with poor load-balance" should be "ESM run with poor load-balance"**

A:

Figure 3 has been updated, improving the text as suggested by the reviewer. We attach the new figure below for reference:

[Figure]

**R8:**

**Couplers like OASIS-MCT support flexible coupling lags which can transform concurrent run among component models into sequential run where one component model will wait in the whole run of another component model. Can the approach in this article be effective for such kind of ESM run?**

A:

Couplers like OASIS-MCT indeed allow for flexible coupling lags, enabling components running on separate processors to execute sequentially, where one component remains idle while waiting for another to complete its timestep. While this approach has been used in the past, it is rarely seen today due to the significant loss of parallel efficiency compared to the concurrent execution mode.

The authors believe that auto-lb would nonetheless work in such scenarios. The core challenge

of load-balancing does not fundamentally change: determining the appropriate parallelisation is key to achieving better parallel efficiency within each coupled component (handled by the prediction script with the individual scalability curves), as well as finding the parallelisation ratio between components to minimise the coupling cost (addressed by the auto-lb workflow).

Even when components run on different physical cores and in sequential mode, increasing the processor count of one component may accelerate its execution and, therefore, reduce the idle time on any component waiting for its fields. However, by doing so it also increases the cost (core-hours) of the accelerated component itself when idle. This tradeoff is still captured by the Fittingness metric proposed, which balances execution speed and resource efficiency.